Sharing refuges on arid islands: ecological and social influence on aggregation behaviour of wall geckos

Vasconcelos Raquel raquel.vasconcelos@cibio.up.pt 1 2
Rocha Sara 3
Santos Xavier 1
1 CIBIO, Centro de Investigação em Biodiversidade e Recursos Genéticos, InBIO Laboratório Associado, Universidade do Porto , Vairão , Portugal
2 IBE, Institute of Evolutionary Biology (CSIC-Universitat Pompeu Fabra) , Barcelona , Spain
3 Departamento de Bioquímica, Genética e Inmunología, Facultad de Biología, Universidad de Vigo , Vigo , Spain
Measey John
Electronic publication date: 2017 Jan 10
Publication date: 2017
Volume: 5
Electronic Location ID: e2802
Received 2016 Aug 11; Accepted 2016 Nov 18
Copyright: ©2017 Vasconcelos et al.
Copyright year: 2017
Copyright holder: Vasconcelos et al.
License: This is an open access article distributed under the terms of the Creative Commons Attribution License, which permits unrestricted use, distribution, reproduction and adaptation in any medium and for any purpose provided that it is properly attributed. For attribution, the original author(s), title, publication source (PeerJ) and either DOI or URL of the article must be cited.
License URL: https://creativecommons.org/licenses/by/4.0/

Keywords: Tarentola substituta, Reptile, Male–female pairs, Thermobiology, Refuge sharing, Density-dependent, Cabo verde

Funding: Fundação para a Ciência e Tecnologia SFRH/BPD/79913/2011 SFRH/BPD/73115/2010 SFRH/BPD/73176/2010 The research and writing were supported by Postdoctoral grants from FCT, Fundação para a Ciência e Tecnologia SFRH/BPD/79913/2011 (to RV), SFRH/BPD/73115/2010 (to SR) and SFRH/BPD/73176/2010 (to XS) financed by The European Social Fund and the Human Potential Operational Programme, POPH/FSE (www.fct.pt). The funders had no role in study design, data collection and analysis, decision to publish, or preparation of the manuscript.

==============================
Background

The extent of social behaviour among reptiles is underappreciated. Two types of aggregations are recognized in lizards: ecological and social, i.e., related to the attraction to a site or to animals of the same species, respectively. As most lizards are territorial, aggregations increase the probability of aggressive interactions among individuals, a density-dependent behaviour.

Methods

After some spurious observations of aggregation behaviour in the endemic Cabo Verde nocturnal gecko Tarentola substituta, we conducted a field-based study in order to thoroughly characterize it. We sampled 48 transects and 40 10 × 10 m quadrats on São Vicente Island to describe the incidence, size and composition of aggregations and to study the effect of gecko and refuge density, plus refuge quality, on refuge sharing. We hypothesize that when density of animals and scarcity of high-quality refuges is higher, lizards have increased probability of aggregating. We also predict a consistent pattern of size and composition of groups (male–female pairs, only one adult male per group) throughout the year if there is a selected behaviour to avoid agonistic interactions, and low thermal advantage to aggregating individuals.

Results

We present one of the first evidences of aggregation for Phyllodactylidae geckos. We found that T. substituta forms aggregations around 30–40% of the time, and that refuges are almost always shared by a female-male pair, sometimes with a juvenile, probably a mechanism to avoid aggressive interactions. We also observed that refuge sharing is dependent on refuge quality, as medium–large (thermally more stable and positively selected) rocks are shared much more frequently than small ones, but independent of adult sizes. Refuge sharing is also directly related to the density of geckos and inversely related to the density of high-quality refuges. We found no relation between body temperatures of geckos and refuge sharing when controlling the effect of rock/air temperature, suggesting that huddling does not improve thermoregulation.

Discussion

Our results suggest that in this harsh environment (rocks reach 46 °C) aggregation incidence is mainly driven by an ecological factor (scarcity of high-quality refuges) and its intersexual composition by social factors (avoidance of agonistic interactions by males, and possible increased reproductive success of the pair). This study sheds some light on the little explored gecko aggregation behaviour and other studies should follow.

Introduction

Many animal species from a wide range of taxa permanently live within groups or form aggregations during certain phases of their life cycle, or under certain environmental conditions, due to beneficial effects (Mouton, 2011). The benefits and costs of living in groups vary from species to species, the latter generally including competition for food resources, reproductive interference and infanticide, inbreeding depression, and higher likelihood of transmission of diseases, parasites and of attracting potential predators with a stronger scent trail (Alexander, 1974; Gardner et al., 2015). Because many species are territorial, as are most lizards, aggregations also increase the likelihood of aggressive interactions among individuals, as agonistic behaviour is density dependent (Wilson, 1975).

In lizards, such as in other animals, two types of aggregations are recognized: ecological and social, i.e., related to the attraction to a site or to conspecifics, respectively (Graves & Duvall, 1995; Kearney et al., 2001). Ecological aggregations typically occur when individuals are attracted to patches of habitat of limited availability or when there is a clumped spatial distribution of exceptionally high-quality factors such as food, shelter, predator avoidance, thermoregulatory or oviposition opportunities (Graves & Duvall, 1995; Kearney et al., 2001). Social aggregations, on the other hand, are formed when individuals are attracted to conspecifics in order to directly benefit from predator avoidance by group defence or increased vigilance, increased thermoregulatory or mating opportunities (Graves & Duvall, 1995; Kearney et al., 2001; Chapple, 2003; Gardner et al., 2015).

In comparison with other vertebrate groups, few references to refuge sharing in reptiles, as well as little quantitative information, exists on the size and composition of these aggregations (Doody, Burghardt & Dinets, 2013; Gardner et al., 2015). Chapple (2003) documented social aggregations in a group of Australian scincids and referred to enhanced vigilance against predators as one of the benefits of group membership. Mouton (2011) reviewed the tendency of five South African lizard species to aggregate in refuges and found that they mostly aggregate due to shortage of optimal shelters and/or for anti-predatory effects. This author argues aggregation behaviour to have resulted in strong male territoriality: adult males attempt to set up harems and only when aggregations become unmanageably big will more than one male occur in the same group (Mouton, 2011). Anecdotal cases of reptile aggregations have been described in agamids (Panov & Zykova, 2003), chameleons (Toxopeus, Krujit & Hillenius, 1988), tropidurids (Vitt, Zani & Caldwell, 1996), sceloporines (Ruby, 1977), xantusiids (Davis et al., 2011), xenosaurids (Lemos-Espinal, Smith & Ballinger, 1997) and amphisbaenidae (Martín et al., 2011), as well as in geckos (Barry, Shanas & Brunton, 2014; Gardner et al., 2015). In geckos, there are apparently no general rules for explaining aggregation patterns and their composition. For example, Meyer & Mouton (2007) showed that Bibron’s gecko (Chondrodactylus bibronii) lives in groups composed by one male and up to 15 females and juveniles, although they could not distinguish whether or not aggregation was the result of limited availability of optimal shelters or of conspecific attraction. Social aggregations which allow the control of thermal exchange rates via huddling were reported for thick-tailed geckos Underwoodisaurus milii (Shah, 2002; Shine et al., 2003) and common brown geckos Woodworthia maculatus (Bauer, 1990). Reduction of evaporative water loss was suggested for the western banded gecko Coleonyx variegatus, which forms large groups under shelters during the day (Lancaster, Wilson & Espinoza, 2006) where often several males and females can be found together (Greenberg, 1943). Pianka & Vitt (2003) suggested the existence of long term pair bond aggregations of the small diurnal naked-toed gecko genus Gymnodactylus in southern Brazil. A link with some life-traits such as viviparity, high longevity and late sexual maturation was also noticed in some Gekkota, and Diplodactylidae (Barry, Shanas & Brunton, 2014; Gardner et al., 2015). These examples demonstrate the existence of few general trends in reptile aggregation.

We aimed to shed some light on the little explored gecko aggregations and the causes of this behaviour. Our study focused on Tarentola substituta Joger, 1984 (Phyllodactylidae), an endemic wall gecko from São Vicente Island, Cabo Verde. This species is a good model to study the effect of density of conspecifics within a species in aggregating behaviour as it is especially abundant in comparison to its relatives in continental Africa and Europe, and to other Cabo Verde endemic geckos (Schleich, 1987; Vasconcelos et al., 2013). This species is strictly nocturnal and carefully selects its diurnal retreat sites (Vasconcelos, Santos & Carretero, 2012). Interestingly, adults tend to select medium–large sized rocks, scarce but high-quality refuges in respect to thermal properties (they reach lower temperatures because they buffer air and soil temperatures better than small rocks, reducing the danger of reaching lethal body temperatures), whereas juveniles used small rocks more frequently (yet still less than expected based on availability alone), sometimes risking overheating (Vasconcelos, Santos & Carretero, 2012). This non-optimal microhabitat choice was attributed to scarcity of high-quality refuges, extremely high densities of conspecifics, and lack of competitors (there is no other reptile species on most of the island, contrary to the rest of the archipelago) and of native nocturnal ground-predators (Vasconcelos, Santos & Carretero, 2012).

In a scenario of high population density, low predation pressure, and scarcity of optimal microhabitats (Vasconcelos, Santos & Carretero, 2012), our study aimed to describe aggregation patterns (incidence, size and composition of groups in age and sex class) of T. substituta on São Vicente Island, and to test if these are dependent on ecological (e.g., quality of the refuge, density of geckos, refuge availability) and/or social factors (e.g., enhanced thermoregulation). Specifically, we have examined whether the tendency to aggregate is mostly related to high-quality refuge scarcity (ecological aggregation) or to improved thermoregulation by physical contact with conspecifics, or increasing mating opportunities (social aggregation). Following the above, we hypothesized that when a lizard population, such as T. substituta, presents high intraspecific density and scarcity of high-quality refuges, has thus increased likelihood of presenting aggregations as predicted by Mouton (2011). Thus, we additionally predict that there will be: (i) an inverse correlation between refuge sharing and availability of high-quality refuges or/and gecko density across sampling sites; (ii) a consistent pattern of size and composition of groups (male–female pairs and only one adult male) throughout the year—based on the assumption that the male-biased dimorphism displayed by the species is an indication of male territoriality—if there is a selected behaviour to avoid agonistic interactions, and not only during the mating season, or a variable pattern along the year if related to mating; (iii) low thermal advantage to aggregating individuals.

Materials and Methods

Study area

São Vicente Island belongs to the windward island group of Cabo Verde Republic (Fig. 1). The island is volcanic with a landscape dominated by stony plains, sandy dunes and barren hills. Except for the summit of Monte Verde (774 m), the island is mostly composed of dry habitats with sparse or no vegetation (Fig. 1). Due to its volcanic nature, almost no burrows or caves are present.

Figure 1 Study area and study site.

Map of the Cabo Verde Islands showing the location of the archipelago relative to the West African coast (A), its elevations and the location of São Vicente Island (B), and of the sampling sites (C), either transects (rectangle) and quadrats (circles); Geographic Coordinate System, Datum WGS84. Mapped habitats (São Vicente) are adapted from Diniz & Matos (1994).

On São Vicente Island, air temperatures are 27.2 ±  0.8°C on average during the warmest months, between July–October, ranging from 24.4 to 28.6°C, and 18.9 ± 0.8°C on average during the coolest months, between December–February, ranging only from 15.7 to 20.4°C due to the moderating influence of the Atlantic Ocean (Hijmans et al., 2005). Air temperature, Ta, is strongly negatively correlated with humidity (Vasconcelos, Santos & Carretero, 2012) and so it can be used as its surrogate.

Study species

Tarentola substituta (Phyllodactylidae) is a flattened, robust oviparous gecko with a long tail and a delicate head with relatively long, pointed snout, and with a mean snout-vent length (SVL) of 51.60 ±  3.64 mm (Joger, 1984; Vasconcelos et al., 2012). Based on previous work, individuals smaller than 45 mm SVL were considered juveniles (Vasconcelos, Santos & Carretero, 2012), as they lacked sexual characters, i.e., ovarian follicles seen by transparency in females, and enlarged hemipeneal spurs and more developed cloacal pouches in males (Vasconcelos et al., 2012). This species also presents marked sexual dimorphism, with adult males presenting, besides the more pronounced cloacal pouches, higher body mass and larger body size than adult females (Atzori et al., 2007; Vasconcelos, Santos & Carretero, 2012). It is distributed all over the island, but avoiding higher elevations, i.e., colder and more humid areas (Schleich, 1987; Vasconcelos et al., 2012). A previous study showed that individuals are active throughout the year (Schleich, 1987). Another study found that the species is hidden in refuges during the day, and that only rocks are used as refuge (Vasconcelos, Santos & Carretero, 2012). No data about the longevity of the species or sister taxa is available.

Geckos and refuge sampling

We have conducted a field-based study to obtain direct evidence of aggregation behaviour. Geckos and refuges were sampled following two complementary methods: (1) linear transects, performed at a single locality with confirmed gecko presence, and (2) quadrats randomly distributed within the main habitats of the island. These two sampling methods had specific and different objectives: transects were used to test whether specific refuges were frequently shared by more than one animal and if some refuges were shared more often than others according to their quality (see below). To test if this refuge sharing was dependent of the densities of geckos or refuge availability, the quadrat methodology was applied. The original raw data are available in Data S1.

Transects were performed on the north-western side of the island, approximately 5 km southwest of the island capital, Mindelo (Fig. 1) in November 2008. During eight days, 48 random transects were performed by two observers in search of geckos under rocks (each 45 min, totalling 36 h of sampling) throughout the diel cycle. Temperatures of body (Tb, of the skin, before touching the animal whenever possible or <10 s after capture), soil under refuge (Ts) and underside of rock used as refuge (Tr) were recorded (in shade if by daylight) with a Fluke® 68 infrared thermometer. Geckos tended to keep their original positions while Tb reading were taken and only one rock was sampled at a time (tilting it to one side) to ensure that readings were made as quickly as possible. Air temperature (Ta, correlated with humidity) was also measured at 10 cm from the ground using a Fluke® 971 temperature-humidity meter (±0.1°C). Thermal readings were made with the thermometer perpendicularly oriented to the surfaces and at very close distances (10–20 cm of the animal) to ensure accurate measurements. The soil type was registered as compact or non-compact because this species avoids the latter (e.g., sandy soils) (Schleich, 1987).

Quadrat sites were randomly chosen stratified by habitat types, with replicates across the entire island according to habitat areas (Fig. 1). The total area and percentage of cover of each habitat registered on São Vicente, the number of quadrats per habitat, and the total and average numbers of adults, juveniles and total number of geckos found in quadrats of each habitat are detailed in Table S1. Sandy and saline areas were not sampled as geckos are absent from such habitats (Schleich, 1987). In total, 40 quadrats of 10 × 10 m were sampled in June 2010. All geckos and refuge rocks available were counted within each quadrat. Rocks stuck to (or into) the ground were not included, as they generally did not offer space for refuge. Sampling time of quadrats varied according to gecko and refuge densities. As T. substituta was found to have exclusively nocturnal activity (Vasconcelos, Santos & Carretero, 2012), sampling of quadrats was performed only during the day to maximize the probability of encountering inactive animals under rocks and obtain the necessary data for calculating densities of geckos per quadrat.

Geckos detected under rocks along transects and quadrats were captured by hand, sexed, photographed (for assigning sex in case of doubt), measured (SVL) for establishing the age category and released at the capture site as quickly as possible to minimize stress. It was registered whenever more than one animal was found under the same rock. Geckos were classified as juveniles, adult males or adult females according to body size and sexual characters. Refuges found to be used by geckos in transects and quadrats were categorised according to rock size. Rock sizes were classified according to their width, length and height by the same observer as small (less than one hand span) or medium–large—(more than one hand span). According to previous work using data-loggers in rocks with different sizes and measuring Ta, Tr and Ts, a clear relation exists between rock sizes and thermal regimes of refuges along the day and Tb (Vasconcelos, Santos & Carretero, 2012). Thus, small rocks were classified as low-quality and medium–large rocks as high-quality refuges for the following analyses.

Statistical procedures

A chi-square test was used to check if the proportion of male–female couples is significantly higher than a random association between pairs of individuals independently of sex. Log-linear analysis was used to test differences in the use of refuges by solitary and pairs of geckos. Specifically we tested the association between three categorical variables in a multidimensional contingency table. The variables were refuge sharing (two classes: shared—whenever two or more geckos used it—and non-shared—when just one individual used it), refuge type (two classes: low and high-quality) and soil type (two classes: compact and non-compact). Log-linear analysis uses a likelihood ratio chi-square statistic. The algorithm generates several models to test interactions among all variables and selects the least complex model that best accounts for the variance in the observed frequencies. Results were interpreted by checking odds-ratio scores in expected values of partial and marginal association tests of variables retained in the model; odd-ratio computes the likelihood ratio statistic of the model containing or not a particular term (significant when p < 0.05) (Jobson, 1992).

Values of Tb were compared between categories of adult geckos (solitary vs. aggregated) with General Linear Models (GLM), using environmental and rock temperatures (Ta and Tr respectively) as continuous predictors (i.e., covariates) to control their effect on Tb, in order to test if refuge sharing could influence Tb of the pair. Only one pair of geckos was found sharing refuges during the night, so we have restricted our analyses to animals found during the day. Juveniles were excluded as refuge sharing was found in very few cases. For geckos sharing a refuge, Tb was considered as the mean value of the two individuals, as Tb was found to be independent of sex in a previous study (Vasconcelos, Santos & Carretero, 2012). In this study, the average difference between Tb of a pair was 1.18°C ±  0.18, and differences in Tb were again not related to sex (Wilcoxon Matched Pair test Z = 1.09, n = 29, p = 0.3).

Non-parametric statistics were used whenever variables distribution violated normality and homoscedasticity assumptions. Mann–Whitney U tests were performed to check whether quadrats with geckos sharing a refuge had a different number of high-quality rocks than those without geckos and to check if adult individuals that shared refuges presented significantly higher SVL than those individuals that did not.

Ethics statement

License nr. 11/2008 was provided by Direcção Geral do Ambiente of Cabo Verde to perform fieldwork in São Vicente that approved all sampling procedures. Geckos detected under rocks along transects and quadrats were captured by hand, sexed, photographed and measured as quickly as possible to minimize stress, and released at the capture site. No animals were killed or harmed. All animals were put in cloth bags during sampling to avoid double counts and to ameliorate stress.

Results

Aggregation incidence, size and composition

In total, 304 geckos were found under rocks (194 in transects and 110 in quadrats). Approximately 32 and 38% of the geckos aggregated under rocks (i.e., were found in contact with or within the distance of their own body length from at least one other individual) in transects and quadrats, respectively. Refuge sharing was detected 50 times (30 times in transects, 20 in quadrats) in a total of 250 sampled refuges (20% of the sampled refuges), 94% of these by a pair of individuals. Refuge sharing was observed between 48 male–female pairs of adults, one male-male pair of adults, and one pair of juveniles. In three cases (two in transects and one in quadrats) a juvenile was found sharing a refuge with a male–female pair. Given that the species has a sex-ratio of near 1:1 (Vasconcelos, Santos & Carretero, 2012), the proportion of male–female pairs is significantly higher than a random association between pairs of individuals independently of sex (chi-square test, χ2 = 25.53; p < 0.001). Adult sizes was not significant for refugee sharing (Mann–Whitney U test, Z = 6057, p = 0.0293; N = 203).

Influence of ecological factors: gecko and refuge densities and refuge quality

The log-linear analysis of frequency tables provided a good adjustment to the best model (maximum likelihood ratio χ2 = 1.30, df = 2, p = 0.52) and retained the interactions between rock size and refuge sharing in the best model (Table 1). The marginal table from which the odds-ratio values were calculated showed that high-quality refuges (medium–large rocks) were shared 5.4 times more frequently than low-quality refuges (small rocks). In contrast, non-shared refuges were selected by single geckos in more similar frequencies, although high-quality refuges were used 1.4 times more than low-quality ones.

Table 1 Results of the log-linear analysis of interaction between refuge sharing, refuge type and soil type.

Results of the log-linear analysis indicating the values of the partial and marginal association tests between the three variables. The informative results come from the interactions between the two environmental variables (refuge quality, RQ and soil type, ST) and refuge sharing, share, with the classes ‘sharing’ and ‘non-sharing.’

	df	Partial χ2	p	Marginal χ2	p	
RQ	1	12.43	0.0004	12.43	0.0004	
ST	1	60.87	<0.0001	60.87	<0.0001	
Share	1	66.42	<0.0001	66.42	<0.0001	
RQ × ST	1	5.700	0.020	7.080	0.008	
RQ × share	1	6.820	0.009	8.190	0.004	
ST × Ref	1	1.170	0.280	2.550	0.110	
Notes.

df degrees of freedom

p p-value

The values of the best model are marked in bold.

A total of 110 geckos were found in quadrats (average density ±  SD = 2.8 ± 3.8 geckos per quadrat). We did not find differences in the densities of geckos across an altitudinal gradient within the commonest arid habitat on the island, i.e., flat, hilly and mountainous (Poisson distribution, log function, Wald statistic W = 4.04, N = 30; p = 0.13). Accordingly, gecko abundances in quadrats with different altitudes were pooled. Geckos were found in 73% of the 40 quadrats examined across the major habitats of the island. Quadrats with geckos (N = 29) had a higher number of high-quality rocks than those without geckos (N = 11) with marginal significance (Mann–Whitney U test, Z = 1.73, p = 0.08; N = 40). Refuge sharing was detected in nine of the quadrats with geckos, seven in arid habitats, and the number of shared refuges increased as more geckos were observed within a quadrat (N = 29, Spearman R = 0.71, p < 0.000; Fig. 2). This test was performed only in arid habitats to avoid a possible bias on densities in other habitat types. The number of high-quality refuges per gecko (average ± SD = 7.84 ± 10.6) was significantly smaller in those quadrats where we found refuge sharing (2.3 ± 1.8) than in those with no refuge sharing (10.2 ± 11.6), even when testing this only for arid habitats (Mann–Whitney U test, Z =  − 2.546, p = 0.001; N = 29).

Figure 2 Relationship between refuge sharing and gecko densities on São Vicente Island.

Refuge sharing is given by the number of shared refuges detected per quadrat and gecko densities by the total number of geckos per quadrat. The Spearman correlation value is also given (R). Values with overlapping points in grey. N, 29 quadrats with geckos (nine with geckos sharing refuges and 20 with single geckos).

Influence of social factors: thermoregulation and mating

Considering diurnal adult individuals observed under refuges, the GLMs did not detect differences between Tb of geckos sharing or not sharing refuges once the effects of Ta and Tr were statistically accounted for (F1,78 = 0.38, p = 0.54). The intersexual composition was observed independently of the time of the year of the sampling, i.e., November and June (R Vasconcelos, 2008, 2010, 2015, X Santos, 2008 & S Rocha, 2010, pers. obs.)

Discussion

This study showed that aggregations are common in T. substituta, with more than a third of the individuals found sharing refuges. It adds to anecdotal observations of aggregations in other Tarentola species, especially from drier regions, although the composition and causes of aggregation in these cases are unknown (JC Brito, GV Antón, C Rato, R Ribeiro & J Teixeira, pers. obs., 2015). Compared to other gecko species, aggregation in T. substituta is less common than in the Duvaucel’s gecko Hoplodactylus duvaucelii, U. milli, and C. bibronii, where its proportion is between 47 and 71% (Lemos-Espinal, Smith & Ballinger, 1997; Shah, 2002; Mouton, 2011; Barry, Shanas & Brunton, 2014). According to Mouton (2011), the nocturnal lifestyle of geckos, minimising predation pressure during activity, and the ground-dwelling nature of some, with individuals moving away from the shelter to forage, minimizing competition for food, are probably the key determinants allowing the occurrence of the aggregation behaviour. Such is the case in T. substituta, which was described as strictly nocturnal (Vasconcelos, Santos & Carretero, 2012) and observed to actively forage for food (R Vasconcelos & X Santos, pers. obs., 2008). But, are these aggregations driven mainly by ecological or social factors?

In a previous study, it was found that this species carefully selected its diurnal retreat sites based on their thermal properties (Vasconcelos, Santos & Carretero, 2012). Given its high density and that its microhabitat choice is constrained mostly by the high diurnal temperatures, we hypothesized that the tendency of individuals to aggregate was mostly related to high-quality refuge scarcity (ecological aggregation) and not improved thermoregulation by physical contact with conspecifics (thermal aggregation, a type of social aggregation). Our analyses showed that even though high-quality refuges were used by both solitary and paired geckos in higher frequencies than low-quality ones, they were shared much more frequently than for the latter. They also showed that quadrats with the presence of geckos had a significantly higher number of rocks than those without geckos. These results indicate that the presence of geckos is associated to the presence of high-quality refuges, similarly to what was found to Egernia striolata skink (Michael, Cunningham & Lindenmayer, 2010). Moreover, the number of shared refuges increased as more geckos were observed within a quadrat, indicating a strong correlation between refuge sharing and density of conspecifics. Relationship between refuge sharing and high-quality refuges availability was inverse, i.e., in those quadrats where we found cases of refuge sharing the number of medium–large rocks per gecko was significantly smaller than in those with no refuge sharing.

The GLMs did not detect differences between body temperatures of geckos sharing or not sharing refuges during the day, once the effects of air and rock temperature were statistically controlled. Thus, Tarentola substituta does not seem to improve thermoregulation by physical contact with conspecifics, contrary to what happens in U. milii (Shah, 2002). Tarentola substituta seems thus to aggregate due to the beneficial thermal regime of high-quality refuges (that are scarce) and not to improve thermoregulation by physical contact. Differences between U. milii and T. substituta can be the result of a difference in body size (the former is larger, approximately 80 mm SVL as adults, and so body temperatures should naturally vary more slowly), as well as environmental conditions (the former is subjected to much more variable thermal regimes than those existing on São Vicente Island, and possibly the adaptive pressure for social thermoregulation is higher). However, the absence of more studies analysing the causes of aggregation in geckos does not allow generalizing patterns.

Interestingly, our study further shows that T. substituta aggregations are mainly intersexual, as the proportion of male–female pairs is significantly higher than a random association between pairs of individuals, similarly to what was found in H. duvaucelii (Barry, Shanas & Brunton, 2014). This suggests that social factors also play a role. There are few examples of gecko intersexual aggregations, but these are usually composed of only one male and one or more females (Chapple, 2003; Mouton, 2011; Barry, Shanas & Brunton, 2014). The most probable explanation of an intersexual aggregation, still to be experimentally tested, might be to avoid agonistic interactions that potentially might result in fatal or debilitating consequences for reptiles (Cooper & Vitt, 1987). Intraspecific agonistic behaviour is common in geckos, with a high level of intraspecific territoriality and associated male-male aggression (Stamps, 1977). In contrast, male–female and female–female aggression are far less frequent (Bolger & Case, 1992; Petren & Case, 1998; Dame & Petren, 2006; Barry, Shanas & Brunton, 2014). Higher levels of agonistic behaviour between males seems to be associated with marked sexual dimorphism, which generally is expressed by longer and heavier bodies, and wider heads in males than females (Bolger & Case, 1992). In São Vicente Island, Tarentola substituta shows a marked sexual dimorphism (Vasconcelos, Rocha & Harris, 2012) and high population densities, a common trend in island lizard populations (Whittaker & Fernández-Palacios, 2007). These, coupled with the scarcity of high-quality refuges, would potentially lead to an increase of male-male agonistic interactions. We speculate that intersexual sharing of thermally-better refuges might be the solution to reduce agonistic interactions, similarly to what has been observed with Peers’ girdled lizard Namazonurus peersi (Mouton, 2011).

Moreover, other factors such as enhanced chances for reproduction could favour intersexual aggregations. For example, male Eublepharis macularius geckos previously housed with females displayed more territorial and courtship behaviour relative to males housed in isolation (Sakata et al., 2002). Thus, it is possible that males of T. substituta prefer to share high-quality refuges with adult females not only to reduce the probability of aggressive agonistic interactions because they are generally less aggressive, but also to increase the probability of mating. That we found this intersexual composition in two different times of the year (November and June and confirmed this with ad hoc observations in February, April and May in different years) supports some stability of aggregations in this species and dilutes the strength of the alternative hypothesis of explain them as fortuitous mating encounters, even though the precise reproductive period of the species is still unknown. A common feature of stable aggregation reported by Gardner et al. (2015) was the occurrence of juveniles with adults of either or both sexes, as we observed in this study. Further support comes from the observation that the juveniles found together with those pairs were not recently born and from other gecko species with similar trends (Barry, Shanas & Brunton, 2014). Those juveniles may be tolerated by adults because they pose no threat to their reproduction or present any inbreeding risk (see Gardner et al., 2015).

Another question that can be raised is why female–female T. substituta couples do not occur more frequently if agonistic interactions are also low? Dame & Petren’s (2006) experiments have proved that the presence of males affected foraging of females: Hemidactylus frenatus males consumed fewer resources and consistently turned their attention towards courtship when females were present. Following the same authors, the reduced foraging effort by males probably leads to increased foraging opportunities for females, who may increase their resource intake to increase provisioning of eggs in the advent of fertilization. It remains to be explored if this applies to T. substituta system as well.

Aggregation behaviour has evolved repeatedly in groups of reptiles that are mainly solitary, and pair- or kin-bonds are a simple form of socialization that may appear under certain circumstances, such as refuge scarcity (Chapple, 2003; Gardner et al., 2015). Cases such as T. substituta, suggest that this behavioural plasticity may ease the evolution of more complex social behaviours in solitary taxa and supports the idea of broader sociality incidence for reptiles (Doody, Burghardt & Dinets, 2013). This is amongst the first documented evidence of aggregation behaviour in geckos of the Phyllodactylidae family, and it suggests differences from the scenario suggested by Gardner et al. (2015). Many questions related to the determinants and consequences of this aggregation behaviour remain unanswered. For example, we do not know whether male–female pairs actually result in higher mating/reproduction success, and whether those pairs are monogamous associations or not. Monogamous or near monogamous mating systems are known in lizards but are poorly documented—they seem to be the case in Tiliqua rugosa (Bull & Baghurst, 1998), Egernia skinks (Chapple, 2003) and Gymnodactylus geckos (Pianka & Vitt, 2003). Reptile monogamy seems to be more common in long-lived species that produce few descendants, have high parental investment and high survivorship of adult and juveniles (Pianka & Vitt, 2003). Unfortunately, basic ecological data are not available for any of the Cape Verde Tarentola species (Schleich, 1987; Vasconcelos et al., 2013). New studies aimed at disentangling basic ecological trends in reptiles from Cabo Verde would be fruitful to understand ecological processes of islands reptiles and to formulate guidelines for their conservation.

Conclusion

In summary, aggregation behaviour in T. substituta is common and seems to be mainly led by ecological (extrinsic) factors, i.e., scarcity of high-quality refuges coupled with high gecko densities. This species seems to aggregate to benefit from the thermal regime of high quality refuges and not to improve thermoregulation by physical contact. Aggregations mostly involve male–female pairs, possibly to avoid agonistic interactions, but perhaps also to increase mating opportunities. Several aspects of the determinants and consequences of this aggregation behaviour could be further characterized, such as aggregation stability.

Supplemental Information

Supplemental Information 1 Abstract in Portuguese (official language of Cabo Verde)

Click here for additional data file.

Table S1 Material details

Total area (in squared meters) and percentage of cover of each habitat registered on São Vicente Island (Diniz & Matos, 1994), number of quadrats per habitat and total and average numbers (±squared deviation, SD) of adults, juveniles and total geckos found in quadrats of each habitat.

Click here for additional data file.

Data S1 Raw databases

Details for quadrats (per gecko and per station) and for transects.

Click here for additional data file.

We thank to Cabeólica SA for logistical support. We thank Gonçalo Cardoso and previous reviewers for helpful suggestions. We thank John Archer for reviewing the English.

Additional Information and Declarations

Competing Interests

Author Contributions

Animal Ethics

Field Study Permissions

Data Availability

The authors declare there are no competing interests.

Raquel Vasconcelos conceived and designed the experiments, performed the experiments, analyzed the data, wrote the paper, prepared figures and/or tables, reviewed drafts of the paper.

Sara Rocha performed the experiments, reviewed drafts of the paper.

Xavier Santos conceived and designed the experiments, analyzed the data, prepared figures and/or tables, reviewed drafts of the paper.

The following information was supplied relating to ethical approvals (i.e., approving body and any reference numbers):

No in vivo experiments were performed. Animals were just measured, sexed and released at the capture site. There is no Institutional Animal Care and Use Committee (IACUC) or ethics committee in neither my research centre nor in Cabo Verde. No specific permissions were required for these locations/activities. The field studies did not involve threatened or protected species. The target species is not protected following international (http://www.iucnredlist.org/details/13152212/0) or national legislation (Boletim Oficial da República de Cabo Verde n°37, I série, Artigo 9°, Anexo II. Decreto-Lei n°3/2003, 30 de Dezembro de 2002. Ministério da Justiça, Praia, Cabo Verde), so no permit is needed to capture or to use it for research purposes. Land accessed is not privately owned or protected.

Geckos detected under rocks along transects and quadrats were captured by hand, sexed, photographed and measured as quick as possible to minimize stress, and released at the capture site. So, no animals were killed or harmed. All animals were put in cloth bags during sampling to avoid double counts and to ameliorate stress.

The following information was supplied relating to field study approvals (i.e., approving body and any reference numbers):

License nr. 11/2008 was provided by Direcção Geral do Ambiente of Cabo Verde to perform fieldwork in S. Vicente that approved all sampling procedures.

The following information was supplied regarding data availability:

The raw data has been supplied as Data S1.

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
