# Peer review of "Sharing refuges on arid islands: ecological and social influence on aggregation behaviour of wall geckos"

_PeerJ, doi:10.7717/peerj.2802_

## Round 0.1 · original submission · Major Revisions

Both reviewers compliment the authors on this study which is of a suitable scientific standard. However, numerous grammatical issues spoil the presentation. Additionally, Reviewer 2 notes that not all relevant literature is included, and some that is included is not correctly cited.

Note that PeerJ requires you to state when personal communications occurred. I question whether São Vicente needs to be reduced to S. Vicente in text and figures.

Table:
The table legend must explain what the variables RQ and ST are, then use only the defined abbreviations in the table.

Figures:
In Figure 1, I cannot resolve which island is São Vicente, please label the parts a, b & c. In Figure 2, please clarify why n does not tally with figures presented in the text. Please review your decision to include the 0, 0 point. Figure 3 is unnecessary

Reviewer 1 ·

Basic reporting

Basic reporting
• The manuscript is characterized by numerous grammatical errors and poor English; the authors are advised to seek professional help to improve the quality of the English.
• In the legend to Figure 2, it is stated that N = 38, but the graph contains only 13 data points, one of which is a zero point. Because some of the data points may lie on top of one another, it is impossible to see how many quadrats contained no geckos or only one gecko. In these two cases it would not be possible to have shared refuges and the question is then whether these data points should be included in the analysis or not? I don't think so. N should indicate the number of quadrats with two or more geckos.
• There is a clear mix up with the legend of Figure 3. The second part of the legend is referring to Fig 2 and not Fig 3. Anyway, I am not convinced that Figure 3 is needed.
• The soil underneath the rock is as much part of the refuge as the rock itself. It is therefore wrong to refer to the rock temperature as the refuge temperature.
• Some of the grammatical errors are listed below, indicated by line number:
28: delete ‘may’ at end of line
44: delete ‘the’ before ‘southern’
49: on aggregating behaviour
50: replace ‘their’ with ‘its’
60: on most of the island
62: a scenario
65: on SV Island
66: replace ‘changes on body temperature’ with ‘enhanced thermoregulation’
68: improved thermoregulation
83: composed of
94: phyllodactylid gecko?
95: with a mean snout-vent length (SVL) of 51.60.......
100: more pronounced cloacal pouches
104-105: sentence needs to be reconstructed
113: whether specific refuges
115: dependent on the densities of geckos or refuge availability
116: raw data are available (elsewhere in the ms plural is used for data)
122: not clear that refuge temperature is rock temperature
128: avoids
132: total number of geckos
136-137: Sampling time of quadrats varied according to
143: quickly
171: only one pair of geckos
179: homoscedasticity
180: sharing a refuge
186: quickly
194-195: Approximately 32% and 38% of geckos aggregated under rocks (i. e., were found in contact with or within the distance of their own body length from at least one other individual
197: 94% of these by
199: in three cases
200: sharing a refuge
201: sex ratio of near 1:1
202: male-female pairs
238: although the composition and causes of aggregation in these cases are unknown
244: forage, minimizing
245: in T. substituta
252: we hypothesized that the tendency of individuals to aggregate was mostly
253: improved thermoregulation
267-268: Tarentola substituta thus does not seem to improve thermoregulation by physical contact with conspecifics
269: write out genus name at beginning of sentence
271: can be the result of a difference in body size
273: as well as environmental conditions
274: than those existing on
276: aggregation in geckos does not
278: male-female pairs
280: but these are are usually composed of
284: with a high level
296: factors such as
297: male Eublepharis macularius
319: socialization that may
323-325: sentence needs to be reconstructed
329: but are poorly documented – they seem to be the case in Tiliqua rugosa
336: processes of island reptiles and to formulate guidelines for their conservation
343: interactions, but perhaps also to
370-71: remove imbedded link
378: delete commas between surnames and initials
379: remove comma after volume number
380: delete commas between surnames and initials
405: species name in italics
406: remove imbedded link
407 & 409: Mouton PleFN
447: remove imbedded link

Experimental design

• The research problem is reasonably well-formulated, except for the hypothesis and predictions, which need to be reconsidered. I suggest something like the following:
Hypothesis:
Aggregation behaviour in T. substituta is caused by a scarcity in high quality refuges.
Predictions:
1. There will be an inverse correlation between refuge sharing and availability of high-quality refuges.
2. There will be an inverse correlation between refuge sharing and gecko density across sampling sites.
3. Based on the assumption that the male-biased dimorphism displayed by the species is an indication of male territoriality, aggregations will not contain more than one adult male.
4. There will be no thermal advantage to aggregating individuals.
5. Male-female pairs will be present throughout the year and not only during the mating season.
• The statistical treatment of the data appears to be sound and the methods are described in sufficient detail to allow replication.
• The authors do not provide sufficient information on exactly how refuges were sampled. Were the rocks lifted straight up or tilted to one side, and did the geckos always maintain their original positions until temperature readings were taken?

Validity of the findings

• Sample sizes are sufficiently large.
• The conclusion is a neat summary of the predicted outcomes.

Additional comments

• The study could make a valuable contribution to our knowledge of aggregation behaviour in geckos, provided that the manuscript is tightened up considerably.

·

Basic reporting

Mostly meets standards. Not sure about Figure 3.

Experimental design

Valid and well reported

Validity of the findings

Valid findings

Additional comments

Thanks you for the opportunity to review this interesting paper. The authors outline aggregations in a gecko and as this in unusual, the work contributes to a growing body of research on aggregations in lizards. Generally the manuscript is well written and I have made some specific comments below to improve specific aspects.
I would also suggest that the author could look at the size of lizards within aggregations and solitary – do larger animals aggregate more than smaller ones?
Is figure 3 correct? It does not appear to be the same figure as in the figure legend.

Edits:
Lines 22-24. Change “which probably benefit of enhanced” to “which probably benefit from enhanced”
However I think the paper by Chapple is not correctly referenced. Although protection from predators and thermoregulation is a potential benefit of the group living Egernia, the review did not come to that conclusion.
The Chapple review dealt with a specific lineage of sinks, the Egernia group and not all scincids as is suggested in the text. Also 23 species in aggregations were mentioned not 33.
Another paper Gardner et al. 2015, which the authors mention, dealt with aggregations in a comprehensive way and outlined several additional characteristics of group living based on social aspects such as viviparity, longevity, kin benefits – but these don’t get a mention in the text.
There is another species of gecko’s that have also been reported to live in groups Hoplodactylus duvaucelii, various papers by Barry et al. which should be mentioned e.g Barry M., Shanas U. & Brunton D.H. (2014) Year-Round Mixed-Age Shelter Aggregations in Duvaucel's Geckos (Hoplodactylus duvaucelii). Herpetologica 70, 395-406.

Lines 70/71/72. I am not sure the sentence makes sense? Perhaps the word “presents” should be “occurs in” and again “presenting should be “forming”
Following the above, we hypothesized that when a lizard population, such
as T. substituta, presents higher intraspecific density and scarcity of high-quality refuges, has
thus increased likelihood of presenting aggregations as predicted by Mouton (2011).
Lines 93-106 Study species: it would be useful to have an indication of the longevity of this species.
There has been some interesting work looking at crevice availability and lizard aggregation by Michael et al and this should be mentioned in the introduction or discussion. Michael D.R., Cunningham R.B. & Lindenmayer D.B. (2010) The social elite: Habitat heterogeneity, complexity and quality in granite inselbergs influence patterns of aggregation in Egernia striolata (Lygosominae: Scincidae). Austral Ecology 35, 862-70.

Line 197/198 change “94% of those times by a pair of individuals” to “94% of those times was a pair of individuals” or perhaps “94% of those times a pair of individuals was detected”

Lines 280-281 There are many more intrasexual observations. For example Tiliqua rugosa pair up with the same mate year after year (e.g. Bull 1988). Bull C.M. (1988) Mate Fidelity in an Australian Lizard Trachydosaurus-Rugosus. Behavioral Ecology and Sociobiology 23, 45-9. See also Gardner et al. 2015.

It would be useful to also highlight at the end of the manuscript that knowing if these aggregations are stable over time – different seasons and between years in the same season, would be of benefit.

---

## Round 0.2 · accepted · Accept

I am happy with the revisions made, and thank you for doing a thorough job. Personally, I don't agree with your decision to limit your discussion to geckos. There is clearly a lot more to be done and other groups, especially other lizards, can lead the way in terms of formulating future hypotheses to test.